# Mutational Analysis of *c-KIT* and *PDGFRA* in Canine Gastrointestinal Stromal Tumors (GISTs)

**DOI:** 10.3390/vetsci9070376

**Published:** 2022-07-21

**Authors:** Maria Morini, Fabio Gentilini, Maria Elena Turba, Francesca Gobbo, Luciana Mandrioli, Giuliano Bettini

**Affiliations:** 1Department of Veterinary Medical Sciences, University of Bologna, Ozzano dell’Emilia, 40064 Bologna, Italy; fabio.gentilini@unibo.it (F.G.); francesca.gobbo3@unibo.it (F.G.); luciana.mandrioli@unibo.it (L.M.); giuliano.bettini@unibo.it (G.B.); 2Genefast srl, 47122 Forlì, Italy; me.turba@genefast.com

**Keywords:** canine gastrointestinal stromal tumors, *PDGFRA*, *c-KIT*, GIST, *PDGFRA* mutation, *c-KIT* mutation

## Abstract

**Simple Summary:**

Gastrointestinal stromal tumors represent the most common mesenchymal tumor of the canine gastrointestinal tract. Activating mutations in *c-KIT* and *PDGFRA* genes are considered the key molecular drivers of human gastrointestinal stromal tumors pathogenesis and are used to predict the response to Receptor tyrosine kinase inhibitors. However, in veterinary medicine, the significance of these mutations in canine gastrointestinal stromal tumors has not been sufficiently explored yet. The aim of this study is to investigate the mutational status of *c-KIT* and *PDGFRA*, by PCR and sequencing, in 17 canine gastrointestinal stromal tumors. Mutations of *c-KIT* were detected in 47% of cases; in one case, *PDGFRA* mutation was also identified. Although follow-up data were not available for all specimens, based on the information collected, we observed at the time of diagnosis the presence of metastases in cases with *c-KIT* mutation. In conclusion, this study provides evidence for the presence of *c-KIT* and *PDGFRA* mutations in canine gastrointestinal stromal tumors and suggests a potential association of *c-KIT* mutation with the more aggressive biological behavior of the tumor.

**Abstract:**

Gastrointestinal stromal tumors (GISTs) are the most common mesenchymal tumors of the canine gastrointestinal tract and are diagnosed by the immunohistochemical expression of the receptor tyrosine kinase (RTK) KIT. Activating mutations of the proto-oncogenes *c-KIT* and *PDGFRA* drive GIST oncogenesis and are used to predict the response to RTK-inhibitors in human oncology. Currently, the frequency and significance of these mutations in canine GIST have not been adequately explored. Therefore, we investigated the mutational status of *c-**KIT* (exons 9, 11 and 13) and *PDGFRA* (exons 12 and 18) genes by PCR followed by fragment analysis for *c-KIT* deletions and PCR followed by screening with DHPLC and direct sequencing confirmation for single nucleotide variations in 17 formalin-fixed paraffin-embedded canine GISTs confirmed by KIT immunopositivity. *c-KIT* mutations were detected in 47% of cases, with a mutation detection rate significantly higher (*p* = 0.0004, Fisher’s exact test) and always involving exon 11. A *PDGFRA* gene mutation (exon 18) was identified in one case. Even if follow-up data were not available for all cases, four cases with documented abdominal metastases displayed *c-KIT* mutations. These data confirm that *c-KIT* exon 11 mutations occur frequently in canine GISTs, and identify the presence of a *PDGFRA* mutation similar to human GISTs. This study also suggests a potential association of *c-KIT* mutation with more aggressive biological behavior.

## 1. Introduction

Gastrointestinal stromal tumors (GISTs) are the most common mesenchymal neoplasms affecting the gastrointestinal (GI) tract wall in humans [1,2] as well as dogs [3] and arise from the neoplastic transformation of interstitial cells of Cajal (ICC) [1,2,4], which are ‘pacemaker’ cells interposed between the plexuses myenteric and muscular tunic of the GI tract, with the function of coordinating peristalsis. Most GISTs are diagnosed on the basis of the immunohistochemical demonstration of the expression of KIT (CD117, stem cell factor receptor), a type-III tyrosine kinase receptor encoded by the protoncogene *c-KIT*, which, therefore, has evidential value in the diagnosis of GIST, which is otherwise difficult to distinguish from other GI mesenchymal neoplasms. In fact, retrospective studies regarding dogs and humans have shown that most of the gastrointestinal mesenchymal neoplasms, previously classified as leiomyomas or leiomyosarcomas, express CD117 and are then to be diagnosed as GISTs [5,6,7,8,9,10,11,12,13,14,15,16].

Mutations in the proto-oncogene *c-KIT* in exon 11, 9, 13 and 17 of the *c-KIT* gene have been observed in human GISTs. The mutation with the highest frequency (of more than 65%) is located in exon 11, in the juxtamembrane domain [17,18,19]. The frequent finding of these mutations has demonstrated their important pathogenetic role, which is responsible for the constitutive dimerization and activation of the KIT receptor regardless of the interaction with the ligand. In canine GISTs, activating mutations have been found and all of them in exon 11, as recorded in humans [7,8,12,15].

*PDGFRA* encodes a transmembrane type-II tyrosine kinase receptor that is normally activated by platelet-derived growth factors (PDGFs), mitogens for cells of mesenchymal origin [20]. *PDGFRA* mutations were identified in 35% of human GISTs with wild-type *c-KIT* [20] and mostly (about 90%) localized in exon 18. In canine GISTs, only Gregory-Bryson et al. (2010) [12] found the presence of synonymous single nucleotide polymorphism (SNPs) in exon 12 and 14 of *PDGFRA*.

Two Tyrosine kinase inhibitors (TKIs), toceranib and masitinib, have recently entered into veterinary clinical practice [21,22] and are currently changing the therapeutic approach for malignancies in dogs and cats. For the treatment of human GISTs, imatinib mesylate, an anticancer agent targeting several protein tyrosine kinases including *c-KIT* and *PDGFR* [23], is the first-line therapy for GISTs carrying the *c-KIT* [24] and *PDGFR* mutations [20,25]. A favorable response to imatinib mesylate has also been reported in dogs with unresected or metastatic GISTs carrying *c-KIT* mutations, suggesting the possibility of the effectiveness of the treatment [7,15]. Strikingly, a metastatic GIST reported by Irie et al. (2015) [7] showed complete remission after imatinib mesylate treatment. A recent study has shown the biological activity of toceranib in canine GISTs with shorter progression-free intervals in GISTs with metastasis and high tumor mitotic index [26]. Therefore, it is important to identify *c-KIT* and *PDGFR* mutations in canine GISTs so that appropriate therapeutic options can be identified.

In this study, the type and frequency of *c-KIT* exon 8, 9 and 11 and *PDGFRA* exon 12 and 18 mutations, and the potential relationship between mutational status and histotype and biological behavior, were investigated.

## 2. Materials and Methods

### 2.1. Samples Collection

The study was conducted at Department of Veterinary Medical Sciences (DIMEVET) University of Bologna (Ozzano Emilia, Bologna, Italy) and was carried out on formalin-fixed paraffin-embedded histological samples of canine GISTs received at the Pathological Service for histological diagnosis over a period of 16 years (2000–2016). All specimens previously diagnosed as GISTs by histomorphology and CD117 immunopositivity were included in this study. From records, anamnestic information about the caseload, including breed, gender and age of the patient, tumors localization, and the presence of metastasis, when detectable, were extracted.

### 2.2. Histology

The classification was based on histopathological observations according to the criteria established by the WHO specific for the tumors of gastrointestinal tract [3], and on the basis of the veterinary literature [5,6,7,8,9,10,12,13,14,15,16]. During the histological evaluations, the morphological type of each case was identified, and they were classified according to their histological pattern (storiform, epithelioid, myxoid and fascicular). Malignant tumors were classified using criteria based on cell differentiation, presence and extension of necrosis within the neoplastic mass, and mitotic index [27,28]. For the evaluation of the histological grade, the samples were scored from 1 to 3 for each of the three previous parameters: overall cell differentiation (1, neoplasms that show close resemblance to the physiological tissue of the adult animal and do not show cellular atypia; 2, neoplasms showing a defined histological subtype or moderate atypia; 3, neoplasms with poor differentiation and marked atypia); mitotic activity (1, range from 0 to 9 of mitotic figures for ten fields at high magnification; 2, range from 10 to 19 of mitotic figures for ten fields at high magnification; 3, 20 or more mitotic figures for ten fields at high magnification) and necrosis (1, absence of necrosis; 2, necrosis ≤ 50% of the tumor surface of the histological section; 3, necrosis > 50% of the tumor surface of the histological section). Grade I was assigned to a final score of 3 or 4; Grade II was assigned a score of 5 or 6 and Grade III was assigned with a score of 7, 8 or 9.

### 2.3. Immunohistochemistry

All the GISTs were subjected to immunohistochemical (IHC) evaluations with CD117, PDGFRα, SMA and Desmin.

The immunohistochemical staining was performed following the streptavidin-biotin-peroxidase technique (BIO SPA, Milan, Italy). The antibodies and their dilutions are as follows: CD117-protein *c-KIT* (1:500, polyclonal; Dako, Glostrup, Denmark), desmin (1:100, monoclonal; Dako), actin smooth muscle (1:100, monoclonal; Dako), and PDGFRα (1:300, polyclonal; Santa Cruz Biotechnology, Santa Cruz, CA, USA). After incubation with 0.3% hydrogen peroxide in methanol for 20 min (to block the activity of endogenous peroxidases), and treatment in a microwave oven (750 W) for the unmasking of the antigen in citrate buffer at pH 6.0 (one cycle of 5 min and 5 cycles of 2 and a half minutes each, topping up the evaporated buffer after each session), the sections were then incubated overnight at 4 °C in a humid chamber with the primary antibody diluted, according to the appropriate dilutions, in PBS (0.01 M, pH 7.4). The sections, subsequently washed in PBS, were first incubated with the secondary antibody (anti-rabbit IgG conjugated with biotin) for 30 min at room temperature, then incubated with the streptavidin–peroxidase complex for 25 min at room temperature. After a 12-minute passage in the DAB chromogen solution (diaminobenzidine 0.02%, and H_2_O_2_ 0.001% in PBS), the sections were immediately rinsed in PBS, then in running water, stained with a counterstain (hematoxylin; Histo-Line Laboratories, Pantigliate, MI, Italy), dehydrated and mounted with DPX (Fluka, Riedel-de Häen, Germany).

The results were graded as follows: −, negative reaction; +, foci of positivity (<50% of neoplastic cells show a positive reaction); ++, widespread cellular positivity (>50% and <75% of cells immunoreactive); +++, most (>75%) of neoplastic cells are positive. Appropriate positive controls were used in order to evaluate the specificity of the reactions and ascertain the appropriate cross-reactivity in the dog tissue. Normal skeletal muscle and normal dog intestine were used as positive controls for mesenchymal markers (SMA, Desmin and PDGFRα). Purkinje cells from cerebellar tissue and mast cell tumors were used as a positive control for CD117 [29,30]. As a negative control for the immunohistochemical procedure, 10% normal goat serum was used on the replicated sections instead of the primary antibody.

### 2.4. Analysis of c-KIT and PDGFRA Mutations

Tissue sample from each case was analyzed to determine the *c-KIT* and *PDGFRA* mutations status in genomic DNA.

Extraction of the DNA from a 6–8 µm section of formalin-fixed paraffin-embedded tissue was performed using the NucleoSpin^®^ FFPE DNA kit (Macherey-Nagel, Düren, Germany) according to the manufacturer’s instructions. Different primers for *c-KIT* and *PDGFRA* (Table 1) were used for PCR amplification.

The PCR products were subjected to the following post PCR analyses:

Exon 8 *c-KIT*: Genescanning analysis, DHPLC and direct sequencing;

Exon 9 *c-KIT*: DHPLC and direct sequencing;

Exon 11 *c-KIT*: Genescanning and fragment analysis on DHPLC;

Exon 12 *PDGFRA*: DHPLC and direct sequencing;

Exon 18 *PDGFRA*: DHPLC and direct sequencing (see Appendix A for details).

### 2.5. Statistical Analysis

Statistical analysis was performed using the Fisher’s exact test to examine the associations between the presence of *c-KIT* mutations and histotype, histological grade and the presence of metastases.

## 3. Results

### 3.1. Sample Collection

Seventeen cases of canine gastrointestinal stromal tumors were included in this study. Nine of these patients were males and 6 were females, and two were of an unknown gender; half of them were sterilized (Table 2). Two of the affected dogs were German Shepherd (2/17; 12%) and two Setter (2/17; 12%), while all the other breeds are represented with only one case; finally, 5 out of 17 dogs were Mixed breed (29%). The age of onset of the primary neoplastic mass ranged from 5 to 14 years, with a mean age of 11 years (Table 2). The localization of the tumor was at the gastric level in two cases (2/17; 12%), one of which was near the pylorus. In nine subjects (9/17; 53%), the GISTs were found in the small intestine. Of these, three involve the duodenum (30%), two the jejunum, two the ileum and two had unspecified location. The remaining six cases (35%) mainly involved the large intestine with two cases at the level of the ileocecal valve, three at the level of the cecum and one at the colon (Table 1) (Figure 1). Metastatic involvement is reported in 4 out of 17 subjects (24%) affecting the tributary lymph nodes and the mesentery in case #13; liver, diaphragm and mesentery in case #7; spleen in case #9 and mesentery in case #10 (Table 2).

### 3.2. Histology

Four out of seventeen GISTs (24%) showed high cellularity, with a storiform pattern. Upon histological observation, they appear highly cellular, consisting of densely packed sheets of cells, with a markedly spindle morphology, arranged in palisades or intertwined and/or intercalated bundles or structured in a ‘storiform’ arrangement, with shaped nuclei oval and eosinophilic cytoplasm. The cell limits appear indistinct (Figure 2a). In three cases (#2, 6, 16) it was possible to observe an abundant presence of necrosis (>50%), hemorrhagic foci, marked neoangiogenesis and extremely dilated and blood-filled blood vessels.

In 7 cases out of 17 (41%), the neoplasm shows a fasciculate pattern with less marked cellularity than the previous ones, and on histological observation it appears to consist of cells with fusiform morphology, which are less densely packed and arranged in loosely intertwined palisades and of smaller extension (Figure 2b).

In 5 out of 17 samples (29%), the neoplastic cells showed an epithelioid-like appearance, and appear arranged in trabecular-like structures or, more frequently, in a solid-compact carpet. The cells, with a polygonal to rounded shape, have an often-vacuolated cytoplasm and high pleomorphism (Figure 2c).

In one case (#12), it is possible to observe a myxoid pattern, characterized by poorly structured cells, spindle-shaped and separated by an abundant myxoid matrix. The nuclei are very elongated with clearly evident nucleoli. Interspersed with these areas it is possible to observe more or less extensive areas with a fascicular or storiform pattern (Figure 2d).

At the evaluation of the histological grade, 10/17 cases (59%) showed a Grade I; 6/17 cases (35%) have Grade II, and only one case (n° 11) is Grade III (Table 2). Evaluation of the presence and extent of intratumoral hemorrhage gave the following results: 7/17 (41%) grade 0, 3/17 (18%) grade 1 and 7/17 (41%) grade 2 (Table 2).

### 3.3. Immunohistochemistry

In all 17 GISTs, the cells show a strong and widespread cytoplasmic immunopositivity of the granular type for the KIT protein (CD117) with a percentage of immunopositive cells greater than about 90%. In only one case (#4) immunoreactivity occurs mainly in the cell membrane (Figure 3a) and in another (#8) immunoreactivity aggregated at the perinuclear area with focal distribution (Figure 3b).

Ten samples were tested with SMA antibody; only one of these (#6) was found to be diffusely immunopositive (Figure 3c), while the remaining nine cases showed focal and weak immunopositivity, with a predominantly cytoplasmic pattern.

Eight samples were tested for Desmin antibody. Of these, seven were negative, while only one sample (#1) was focally positive at the cytoplasmic level, in about 30% of the cells.

Of all samples tested for PDGFRα antibody, three cases were completely negative, eight samples were weakly positive and six samples moderately positive. The immunopositivity pattern for this marker is localized at the cytoplasmic level with variable intensity, from weak to moderate, with a percentage of positive cells greater than 70% (Figure 3d).

### 3.4. Analysis of c-KIT and PDGFRA Mutations

*c-KIT* mutations were detected in 8/17 cases (47%) and always involved exon 11 (deletion of 3–46 bp), while exons 8 and 9 were wild-type in all cases. *PDGFRA* gene mutation was identified only in one case in the exon 18 (ENSCAFT00000003270.5 c.2597 G > A, SNP activating mutation) (Table 3). The mutation in the dog results in the substitution of Arginine with Lysine in position 866 and it was identified by DHPLC screening and confirmed by sequencing, which allowed definition of the mutation. The mutation is predicted to be cancer causative with a score of −2.10 (standard threshold at −0.75) using FATHMM [31].

Although follow-up information was not available for all cases, abdominal metastases were documented in four cases, and all showed a mutation in *c-KIT* exon-11.

### 3.5. Statistical Analysis

No significant correlations were found between the presence of metastases and histological pattern (*p* = 0.2605), histological grade and histotype (*p* = 1.0000), histological grade and presence of mutations (*p* = 0.5840). The presence of mutations, on the other hand, appears to be significantly (*p* < 0.05) correlated with the presence of metastases (*p* = 0.0004).

## 4. Discussion

Gastrointestinal stromal tumors (GISTs) represent a distinctive diagnostic neoplastic entity that, although quite uncommon and not yet fully characterized in dogs, has acquired over the years a well-defined clinical and histopathological identity and generates considerable interest for several reasons. Indeed, canine GISTs bear a very close resemblance to their human counterparts, representing an excellent spontaneous comparative model. This similarity paves the way for significant therapeutic perspectives in dogs, since features and the mutational aspect are almost completely superimposable [6,7,8,12,19,32,33,34,35].

In our caseload, collected over a long period (16 years), GISTs are scarcely represented. However, these data follow epidemiological data from the literature [5,6,10,11,12,13,14,16], which considers GISTs a neoplasm whose incidence is rather difficult to determine but still relatively rare, although it represents an important share of gastrointestinal mesenchymal neoplasms (from 39% to 50%).

CD117 positivity in over 70% of neoplastic cells is the prerequisite for diagnosing these neoplasms as GISTs [3]. In all our tumors, CD117 revealed an intense and widespread positivity to over 90% of the neoplastic cells. For most of the cases, this positivity was of a finely granular type, and in one case a paranuclear positivity was expressed in the form of small rounded and intensely colored clusters near the nucleus. This result reflects literature data [16] in which paranuclear positivity has been reported in a small number of cases.

Some attempts have been made to correlate the CD117 immunohistochemical pattern to the presence of mutations in canine GISTs, similar to what has already been extensively analyzed for mast cell tumors in dogs [36], but without significant results [8]. Furthermore, in our cases, no association was found between the IHC pattern and the mutational status of *c-KIT*.

Eight distinct mutations of the same type (deletions) were identified in this study, all involving exon 11 of the *c-KIT* gene in the juxtamembrane domain. On the other hand, no mutation was recognized in the other exons investigated (exons 8, 9, 13 and 17). These results confirm what has already been reported in canine cases [6,7,12] in which the mutations are localized in exon 11 in almost all of them. Only Irie et al. (2015) [7] reports a mutation in exon 9 of *c-KIT*, which is found in humans, but not in our series.

Almost half of the cases analyzed have this KIT mutation (47%; 8/17), in a percentage that falls within the range of the few studies that have evaluated the frequency of this mutation in canine GISTs (from 35% by Gregory-Bryson et al., 2010 [12] to 50% by Kumagai et al., 2003 [9] up to 74% by Takanosu et al., 2016 [8]). The different results might be the consequence of the employment of different methodological approaches (conventional PCR vs RT-PCR). The presence of KIT mutations does not appear to be related either to the GISTs localization, nor to the histotype and to the histological grade of the GI tumors. Nonetheless, most cases with mutations showed metastases at the time of diagnosis and, thus, exhibited a more biologically aggressive behavior (*p* = 0.0004).

One case showed the presence of an activating point mutation at the level of exon 18 of the *PDGFRA* oncogene (case #4) in correspondence to a human equivalent mutational hotspot. Until now, no significant activating mutation affecting *PDGFRA* had ever been reported in the dog, unlike what had been found for some time in humans, in which activating mutations of this gene represent a share ranging from 10 to 35% [16,19,34]. Interestingly, case #4 also presented an activating mutation in *KIT* Exon 11. This finding, both mutations in the same tumor, has never been observed in human GISTs.

In humans, *PDGFRA*-mutated GISTs are frequently found in the stomach, presented an epithelioid histotype and are often associated with a PDGFRα strong immunopositivity [37]. Of note is that case # 4, although localized in the intestine and with a moderate PDGFRα immunopositivity, shows the same epithelioid histotype.

There are several ‘inhibitory’ molecules that target tyrosine kinase receptors (TKIs), including imatinib mesylate, which has long been successfully used in the treatment of human GISTs with KIT mutations. To date, due to its effective therapeutic response, this TKI has recently been considered the treatment of choice for patients with advanced stage GISTs. In a study conducted on two dogs with documented exon-11 *c-KIT* mutations, imatinib mesylate (Gleevec) has been used for its ability to inhibit protein kinases and results in a marked tumor response and an increase in survival time [7,15]. The use of toceranib phosphate in seven dogs with GIST had recently shown a shorter disease-free interval in dogs with metastasis at diagnosis and a high tumor mitotic index [26]. Therefore, the role of *c-KIT* and *PDGRFA* mutations in canine GISTs must be extensively studied to understand their potential therapeutic efficacy.

## 5. Conclusions

*c-KIT* mutations were detected in half of the cases included in this study, and their presence appears to be associated with malignancy. An activating mutation of *PDGFRA*, which in humans is represented in a significant fraction of GISTs, had not yet been reported in canine GISTs. Unfortunately, follow-up data were not available and, therefore, no meaningful prognostic information can be provided from this study.

In conclusion, the study confirms that activating mutations of *c-KIT* play an important role in canine GISTs, are related to a malignant biological behavior and, therefore, have strong therapeutic implications. Further studies are needed involving a larger number of cases with follow-up data in order to demonstrate the prognostic and predictive role of *KIT* or *PDGFRA* mutations in canine GISTs.

Our results indicate, confirming what has already been stated in the literature, to what extent KIT mutations can have a prognostic and therapeutic significance in canine GISTs.

## Figures and Tables

**Figure 1 vetsci-09-00376-f001:**
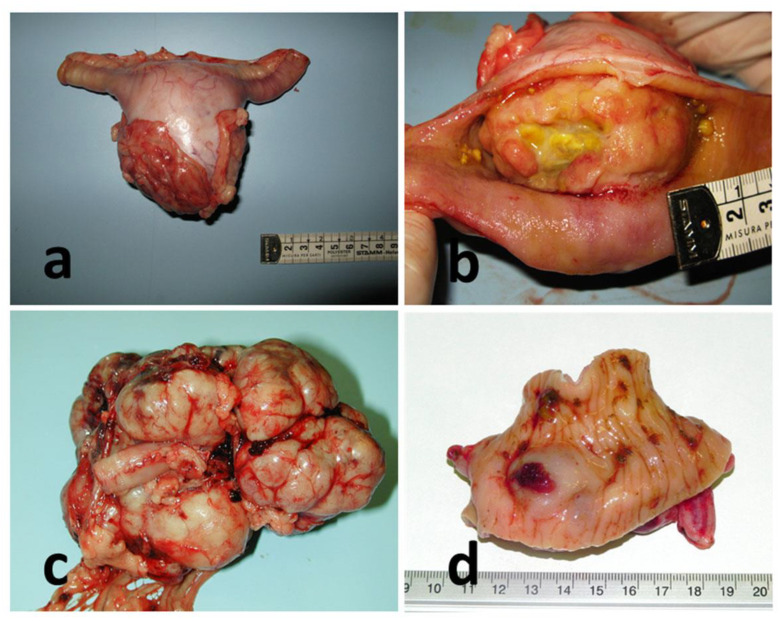
Gross features of GISTs cases. (**a**) Case #11, intestine, jejunum. (**b**) Case #14, intestine, duodenum. (**c**) Case #10, intestine, ileum; (**d**) Case #16, intestine, cecum.

**Figure 2 vetsci-09-00376-f002:**
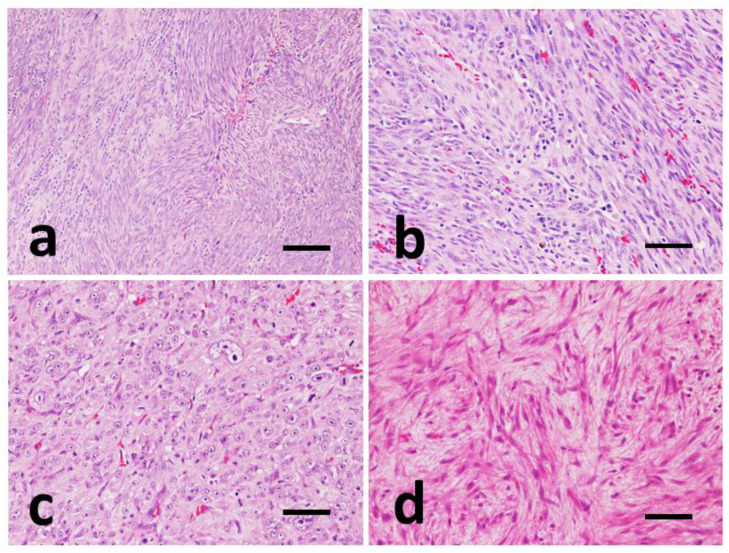
Representative images of histomorphological pattern of GISTs, Hematoxylin-Eosin (HE). (**a**) Case #8, storiform pattern. Spindle cells forming whirls and palisades. (**b**) Case #16, fascicular pattern. Spindle cells are organized into interlacing and crisscrossing bundles. (**c**) Case #9, epithelioid pattern. Nests and sheets of round to polygonal cells. (**d**) Case #12, myxoid pattern. Scattered, non-cohesive, spindle cells with myxoid matrix. (**a**) Scale bar 300 µm and (**b**–**d**) scale bar 100 µm.

**Figure 3 vetsci-09-00376-f003:**
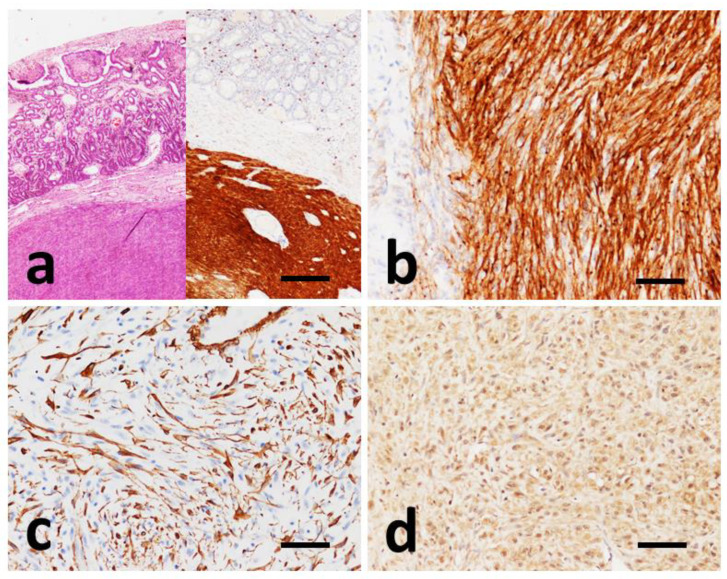
Immunohistochemical findings in GISTs. (**a**) Case #6, HE and IHC with CD117 of consecutive slides showing strong and uniform cytoplasmic CD117 positivity of the tumor mass. (**b**) Case #8, prevalent strong perinuclear dots and membranous staining of CD117. (**c**) Case #6, SMA positivity of neoplastic cells. (**d**) Case #4, cytoplasmic positivity to PDGFRα. (**a**) Scale bar 500 µm, (**b**–**d**) scale bar 100 µm.

**Table 1 vetsci-09-00376-t001:** Primers and their respective annealing temperature used in the study.

Name	Sequence 5′ → 3′	Annealing T
Ex 8 *c-KIT* forward	[Hex]–GGGGAGCCTTGGTGAGGTGT	58 °C
Ex 8 *c-KIT* reverse	CCCTGCTGTCCTTCCCTCGT	58 °C
Ex 9 *c-KIT* forward	ACTCGTCTCTGTCACCGTCTGGAA	58 °C
Ex 9 *c-KIT* reverse	ATGGCAGGCAGAGCCTAAACATCC	58 °C
Ex 11 *c-KIT* forward	[6FAM] ATGATCTGTCTCTCTTTTCTCCCC	60 °C
Ex 11 *c-KIT* reverse	GTACACAAAAAGGTTACATGGAAAGC	60 °C
F_*PDGFRA*_ex 18	GTTCCTTCCCTTTCCATGCA	56 °C
R_*PDGFRA*_ex 18	GTGAGGAAAGGTGGGCTTGTC	56 °C
F_*PDGFRA*_ex 12	TGCGTCTGGGCTTTGATAATT	56 °C
R_*PDGFRA*_ex 12	GATCACCCCAGTAGGCGCTTA	56 °C

**Table 2 vetsci-09-00376-t002:** Anamnestic information and histological results of the 17 canine GISTs. Abbreviations: F: female; FS: spayed female; M: male; MN: neutered male; u: unknown; y: years; a: absent (at the time of diagnosis).

Case	Breed	Sex	Age, y	Location	Tumor Size (cm)	Metastasis	Histotype (WHO) [3]	Grade *	CD117 **	PDGFR **	SMA **	DES **
1	English Setter	M	9	Duodenum	u	a	Fascicular	II	+++	−	++	−
2	Collie	F	9	Duodenum	u	a	Storiform	I	+++	−	−	−
3	German Shepherd	F	12	Ileociecocolic junction	u	a	Storiform	I	+++	+	++	−
4	Mixed	u	u	Cecum	4	a	Epithelioid	I	+++	++	++	−
5	Great Dane	F	10	Small intestine	u	a	Epithelioid	I	+++	++	++	−
6	Mixed	M	11	Small intestine	3	a	Storiform	I	+++	+	+++	−
7	Pekingese	NM	14	Ileociecocolic junction	1	Diaphragm, liver, mesentery	Fascicular	I	+++	−	++	−
8	Poodle	M	13	Stomach	10	Spleen	Storiform	I	+++	+	−	−
9	English Setter	u	7	Stomach	u	a	Epithelioid	II	+++	+	++	−
10	Maltese dog	M	12	Ileum	8	Mesentery	Fascicular	II	+++	++	−	−
11	Schnauzer toy	FS	14	Jejunum	7 × 8	a	Epithelioid	III	+++	+	−	−
12	Mixed	NM	12	Colon	u	a	Mixoid	I	+++	+	−	−
13	Belgian Shepherd	FS	14	Ileum	10	Mesentery, lymph node	Fascicular	I	+++	++	++	−
14	German Shepherd	M	11	Duodenum	3 × 4	a	Epithelioid	II	+++	+	−	−
15	Corso	FS	5	Jejunum	5.7 × 2	a	Fascicular	II	+++	++	++	−
16	Mixed	M	10	Cecum	5 × 2	a	Fascicular	II	+++	+	++	−
17	Mixed	NM	12	Cecum	3 × 1 × 2	a	Fascicular	I	+++	++	−	−

* histological grade, from I to III, were assigned to all samples according to the following criteria: overall cell differentiation (1, neoplasms that show close resemblance to the physiological tissue of the adult animal and do not show cellular atypia; 2, neoplasms showing a defined histological subtype or moderate atypia; 3, neoplasms with poor differentiation and marked atypia); mitotic activity (1, range from 0 to 9 of mitotic figures for ten fields at high magnification; 2, range from 10 to 19 of mitotic figures for ten fields at high magnification; 3, 20 or more mitotic figures for ten fields at high magnification) and necrosis (1, absence of necrosis; 2, necrosis ≤ 50% of the tumor surface of the histological section; 3, necrosis > 50% of the tumor surface of the histological section). Grade I was assigned to a final score of 3 or 4; Grade II was assigned a score of 5 or 6 and Grade III was assigned with a score of 7, 8 or 9. ** Immunopositivity was graded as follows: −, negative reaction; +, foci of positivity (<50% of neoplastic cells show a positive reaction); ++, widespread cellular positivity (>50% and <75% of cells immunoreactive); +++, most (>75%) of neoplastic cells are positive.

**Table 3 vetsci-09-00376-t003:** *c-KIT* and *PDGFRA* gene status of the 17 canine GISTs.

Case	*c-KIT*		*PDGFRA*	
1	Wild-type		Wild-type	
2	Wild-type		Wild-type	
3	Exon 11, deletion	6 bp	Wild-type	
4	Exon 11, deletion	15 bp	Exon 18, SNP	c.2597G > A; p.866 Arg > Lys
5	Wild-type		Wild-type	
6	Wild-type		Wild-type	
7	Exon 11, deletion	3 bp	Wild-type	
8	Exon 11, deletion	6 bp	Wild-type	
9	Wild-type		Wild-type	
10	Exon 11, deletion	6 bp	Wild-type	
11	Wild-type		Wild-type	
12	Exon 11, deletion	48/49 bp	Wild-type	
13	Exon 11, deletion	21 bp	Wild-type	
14	Wild-type		Wild-type	
15	Wild-type		Wild-type	
16	Exon 11, deletion	21 bp	Wild-type	
17	Wild-type		Wild-type	

## Data Availability

The data presented in this study are available on request from the corresponding author.

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
