# Peer review of "Mutational Analysis of c-KIT and PDGFRA in Canine Gastrointestinal Stromal Tumors (GISTs)"

_vetsci, 2022, doi:10.3390/vetsci9070376_

Round 1
Reviewer 1 Report
In this manuscript, the authors have properly investigated the mutational status of c-KIT and PDGFRA genes by PCR in 17 formalin fixed paraffin-embedded canine GISTs confirmed by KIT immunopositivity. Additionally, they have studied the histological subtypes and the histological grade of malignancy. Besides, all the GISTs were subjected to immunohistochemical evaluations with CD117, PDGFRα, SMA and Desmin.
I find it quite interesting, and it adds valuable information to take into account for future studies.Given the low incidence of this special type of gastrointestinal tumor,the number of cases seems adequate to me.
This manuscript is well structured and synthesized; however, I am missing important information about the patients and some of the results have not been adequately discussed.
Therefore, a few issues need to be addressed to ensure that the correct ideas are conveyed:
1. Introduction and design of the study: In the introduction you state: “Most GISTs are diagnosed on the basis of the immunohistochemical demonstration of the expression of KIT” (line 48), and in the material and methods section it is specified that “All specimens previously diagnosed as GISTs by histomorphology and KIT immunopositivity were included in this study.” (Line 90-92). However, in humans, the presence of discovered-on-GIST 1 (DOG1), has been determined to ubiquitously define GIST, regardless of c-Kit expression. Although previous publications have defined GISTs in dogs as mesenchymal intestinal neoplasms expressing c-Kit, recent literature suggests that a combination of c-Kit and DOG1 IHC is most sensitive for diagnosis of GISTs in dogs (Reference: Dailey DD, Ehrhart E, Duval DL, et al. DOG1 is a sensitive and specific immunohistochemical marker for diagnosis of canine gastrointestinal stromal tumors. J Vet Diagn Invest. 2015;27(3):268-277.) Given that the combined use of KIT and DOG‐1 increases the sensitivity of IHC in the diagnosis of GISTs, how do you explain why this marker was not included for the diagnosis of GISTs in this study?
2. Introduction: Line 61-62: “In canine GISTs, activating mutations have been found in a rather high percentage of GISTs, and all of them in exon 11, as recorded in humans” You include here 4 references: references 7 and 15 only include 1 case, in reference 12 the mutation was detected in only one out of twelve cases. Only in the reference 8, the mutation was detected in 73% of cases using RT-PCR. I would not say “rather high percentage” with these references.
3. Results: Line 177-179: “Metastatic involvement is reported in 4 out of 17 subjects (24%) affecting the tributary lymph nodes and the mesentery in case #13, liver, diaphragm and mesentery in case #7, spleen in case #9 and mesentery in case #10 (Table 2).” How was this metastatic involvement diagnosed? Is there any possibility of establishing clinical staging at the time of diagnosis?
4. Results: Can you provide a more detailed macroscopic description of the tumors such as tumor size or ulceration?
5. Results: Table 2: please specify how the grade has been determined and the criteria for +/++/+++. In the histotype, please indicate the reference. Here, in cases #2 and #7, in the PDG column, the symbols are unknown.
6. Results: Figure 1: the images appear blurry in the manuscript, not in the tiff format. Figure 2: the images appear blurry in the manuscript, in the tiff format only a and d are blurred
7. Discussion: Line 306: this result has been previously explained, please do not repeat in the discussion section
8. Discussion: It is widely accepted that none of the GISTs are positive for desmin (Reference: Meuten D., Tumors in domestic Animals). Desmin is a marker which is normally used to confirm smooth muscle origin of sarcomas, therefore, how do you explain your positive case?
Finally, many format issues must be corrected. Please, see the instructions for authors section to adapt them.
Reviewer 2 Report
Dear Authors,
in my opinion, this is a well-written manuscript with a few mistakes.
1. In the Abstract and Introduction You write that GIST is the most commonly diagnosed tumor of the gastrointestinal tract in dogs. But in line 260: "..GISTs are relatively very few represented..." and "...which considers GISTs a neoplasm whose incidence is rather difficult to determine but still relatively rare. Which sentence is true?
2. In line 154 - the part Sample collection, You write: " Seventeen cases of canine gastrointestinal stromal tumors were included in this study. Nine of these patients are males and 6 females, half of them sterilized." Nine plus six is fifteen, not seventeen? Yes, I know the explanation is in table 2 -the sex of the two dogs is unknown. But is a lack of this info in the text. In my opinion, this fragment about the sex of the dogs must be entered in a part Material and methods, sample collection.
3. Figure 2 - scale bar for a b,c,d is 100 um. In my opinion, the magnification of part c is different than parts b and d. I can be wrong, but the cells in part c are bigger than in part b. Please check.
Regards
